


# Gaia: Complex systems prediction for time to adapt to climate shocks

Allen G. Hunt[1] Muhammad Sahimi[2], Boris Faybishenko[3], Markus Egli[4], Zbigniew J. Kabala[5], Behzad Ghanbarian[6], and Fang Yu[7]

[1]Department of Physics, Wright State University, 3640 Colonel Glenn Highway, Dayton, OH 45435 USA

[2]Department of Chemical Engineering and Materials Science, University of Southern California, Los Angeles, CA, USA

[3]Energy Geosciences Division, Lawrence Berkeley National Laboratory, University of California, Berkeley, CA 94720, USA

[4]Department of Geography, University of Zurich, Winterthurerstrasse 190, CH-8057 Zurich, Switzerland

[5]Department of Civil & Environmental Engineering, Duke University, Durham, NC 27708 USA

[6]Porous Media Research Lab, Department of Geology, Kansas State University, Manhattan, KS 66506 USA

[7]Department of Forestry, Beihua University, 3999 Binjiangdong Road, Jilin, China, 132013

*Correspondence to*: Allen G. Hunt (allen.hunt@wright.edu)

**Abstract.** Earth's climate has undergone significant fluctuations in the geologic past. We focus on the glacial episodes that followed the major waves of invasion of land plants. Twice in Earth's history the impacts of land plant innovations on the atmosphere through increased $CO_2$ drawdown have precipitated sufficient cooling to produce ice ages. Each time, however, adaptation of soil ecosystems eventually helped re-establish apparent steady-state conditions, i.e., new equilibrium temperature and atmospheric $CO_2$ content, ending each of the glacial episodes. In each case, the time interval between the initial innovation and the emergence from the glacial episode was approximately 60 Myr. The consistency of the time scale of the response invites an explanation in terms of a universal rate of dispersal of genetic information that encapsulates the biogeochemical cycle of cellulose production and decay. In this paper, we postulate that the long time for adaptation is a consequence of the time required for the spread of an entire clade though the soil to continental scale. Although 60 Myr appears to be a long time, it is very short compared to the time required for diffusion to transport even molecules like $HCO_3^-$ or sugar through the soil over a continental distance of 5,000 km, which is between $10^{14}$ and $10^{16}$ years for solutes with such soil diffusion constants in the range $10^{-11}$ $m^2 s^{-1}$. Horizontal solute transport through heterogeneous media by advection might be considered as a dispersal mechanism, but is also known to require enormous time scales (ca. 150 Myr for 500 m). We also seek a relevant mechanism in the known scaling of plant and fungal growth rates as a function of time, which can facilitate as well the movement of bacteria, and predicted these rates theoretically on the basis of the universal optimal 2D paths tortuosity from percolation. Comparison with actual data pairs (7,000) for plant and fungal growth rates were used to verify these predictions over 13 orders of magnitude of time, from about one minute to 100 kyr; extrapolation over less than three additional orders of time yields a continental-scale transport time of 80 Myr. We now interpret this prediction in terms of Margulis' understanding of emergent behavior of coupled (soil and plant) ecosystems responding to climate shocks induced by plant innovations.



## 1 Introduction

In his classic work *Gaia: A New Look at Life on Earth*, James Lovelock (1979) proposed that Earth's biosphere is a global-scale self-regulating organism. Whether or not such an interpretation is justified, there is certainly significant evidence that

Earth is life-supporting and a self-organizing system. Lovelock and Margulis (1974) emphasized that the concentrations of $CH_4$, $O_2$, and $CO_2$ in Earth's atmosphere are more than 30 orders of magnitude out of their equilibrium values, which is in contrast to the atmospheric chemistry of Earth's nearest-neighbor planets likely obeying chemical equilibrium. Kleidon (2002) and Lenton (2002) referred to bounded fluctuations in Earth's mean temperature over the past ca. half billion years, and these authors indicated that atmospheric and soil moisture conditions, which have been produced by life, increased plant productivity

by 250%. The same authors also pointed to the failure of Earth to warm despite being affected by warming sun. The original Gaia hypothesis has, however, been mostly abandoned (Schneider, 1986, 2002), even though the concept that the biosphere is composed of interacting complex systems exhibiting emergent behavior (Margulis, 1999), is rather generally accepted. Importantly, Doolittle (2017) reversed his own earlier criticism that Darwinian evolution of species contraindicated the concept of Gaia, by invoking the stability of clades adapted to facilitate biogeochemical cycling, despite the replaceability of individual

species within the clades. This shift in focus, from species to clade, invites explanations for the time span required for establishment of a species assemblage over a soil network in terms of transport times within the soil.

In our perspective, biological innovations based on photosynthesis (bacteria and plants) that allow for better exploitation of the atmosphere's carbon pool are viewed as (cooling) shocks to the system. These shocks have caused a storage of solar energy (Kleidon, 2002), while the overall soil ecosystem, including bacteria, animals, and fungi, consumed some of that energy, and

responded by promoting homeostasis (Bromhall, 2019). Therefore, we hypothesize that the global temperature is then regulated not solely by the rate of injection of oxygen into the atmosphere through photosynthesis, but in tandem with the rate at which it is removed through metabolism. In this paper, we propose the fundamental scaling functions to describe soil-based bacterial and fungal adaptation to new plant strategies. This adaptation appears to be slower than the relatively rapid spread of plants across the land through release of spores in the atmosphere, so that the homeostasis is delayed by the need for transport of all

the members of the plant/soil ecosystem mostly through the soil. However, even though plant growth is a very slow process compared to atmospheric spore transport, it is far more rapid than other means of transport through the soil, such as diffusion and advection by flowing water (Hunt et al., 2021; Hunt and Manzoni, 2016). We addressed this in the framework of transport over the physical network that underlies the biological networks existing in the soil and also discussed some criticisms of the Gaia hypothesis.

The research presented in this paper is based on the analysis of statistical mechanics of heterogeneous media (a branch of complex systems research) to predict a time scale for the formation of a continental-scale "organism"). Although already obtained (Hunt and Manzoni, 2016), the potential relevance of this result to Paleozoic ice ages is also proposed. The subsequent discussion resolves some criticisms of the Gaia hypothesis.





Our analysis is based on an existing scaling relationship for plant growth (and fungal hyphae), which was verified (Hunt et al. 2020b, 2022) by comparison with growth of pines (Ryan and Yoder, 1997) and eucalyptuses (Givnish et al., 2012), the former along a precipitation gradient of factor 4 and the latter along an even larger precipitation gradient of a factor 20 (Ryan and Yoder, 1997) as well as on a wide range of root growth rates, tree heights and clone extents over time scales (Hunt, 2017) from about a minute to 100,000 years. The predicted dependences on time and transpiration were demonstrated) to be consistent with the entire BAAD (Biometric and Allometric Data) set (Falster et al., 2015) of over 6,000 data points of vegetation heights (for numerous species) as a function of age. The time dependence was also verified independently for over 30 different species of tree. The vast time scales addressed connect pre-agricultural with present times, and cover more than 13 of the 16 orders of magnitude of time scale that connect the pore to the continental scales. The transpiration dependence allowed for formulation of the water balance in terms of an ecological optimum (Hunt et al., 2023). The generality (it works for fungi), also versatility, theoretical basis, scope, and accuracy of the scaling relationship justify its proposed use (Hunt and Manzoni, 2016) to "extrapolate to a time scale when an 'organism' with an optimal, hierarchical, structure would reach continental size (about 5,000 km, if growing from the center). That time scale is less than 100 Myr (about 80 Myr)." We show here that 80 Myr is very nearly an appropriate time span in the context of restoration of the Earth to higher temperatures subsequent to Paleozoic ice ages triggered by colonization of the land by plants.

## 2 Theory

In any integrative sense, the existence of a global-, or at least continental-scale "organism" (think interacting ecosystems (Margulis, 1999)) requires some means of communication over such a scale (Hunt and Manzoni, 2016). If this communication is tied to the growth of plants and fungi, the assessment of controlled homeostasis should relate to the prediction for increase in root lateral spread (RLS) $l$ of plants/fungi as a function of time and water flow rates in the subsurface. Thus it was proposed that a continental-scale "organism" could consist of a synergistically operating system of plant roots and associated bacteria. Given that fungi follow the same growth model as plants and act to decompose wood (Bromhall, 2019), they may be included as well. In the case of bacteria, the particular means of communication is not specified, whether chemical, genetic, or otherwise. Bacteria have been shown to act collectively (Lyon et al. 2021), e.g., by deferring consumption until better food sources are located, communicating (Chimileski et al., 2017) at relatively short length scales through chemical "quorum sensing," and at longer length scales through electrical signaling along ion channels (Beagle and Lockless, 2015). Overall, we appeal to the fundamental thesis of Hunt and Manzoni (2016), that the physical transport times over soil networks limit the biological communication times under any proposed scenario.

Within the soil, plant roots are likely to follow paths of minimal cumulative resistance and soil flow paths can be described using models of fractal geometry (Hunt and Sahimi, 2017a; Porto et al., 1998). The RLS relates to the root length $L$, through a non-linear power-law, $L \approx l^{\delta}$. The exponent $\delta$ is known as the fractal dimension of the optimal paths in heterogeneous media,



which reflects confinement of plant root ecosystems globally to a very thin soil layer, with a universal value of 1.21 (Hunt and
      Sahimi, 2017a; Porto et al., 1998).

      Predictions for the RLS (for the case of "biological transport") of plants can be made using the equation given by (Hunt et al.,
      2020):

$$x = x_0 \left(\frac{t}{t_0}\right)^{\frac{1}{D_{opt}}} \tag{1}$$

where $x_0$ is a fundamental pore separation (or xylem diameter) and $t_0$ is the time required for water to traverse a pore separation
      and $D_{opt} = 1.21$, which is the optimal paths exponent from percolation theory (Porto et al., 1998), and valid when the paths are
      confined to a 2D layer, such as the soil layer. In Eq. (1) we identify the RLS, $l$, with, $x$, and the ratio $t/t_0$ as the number of pores
      the root has grown through, which corresponds to root length $L$ in multiples of $x_0$. The specific form of Eq. (1) is chosen to
      emphasize the similarities between scaling functions for plant growth and soil formation (discussed below) as well as the
associated mechanisms potentially available for intra-clade communication through the soil.

      For soil depth predictions (for the case of "physical transport"), one may use the equation given by (Hunt and Sahimi, 2017a;
      Egli et al., 2018; Yu et al., 2019; Yu and Hunt, 2017ab)

$$x = x_0 \left(\frac{t}{t_0}\right)^{\frac{1}{D_b}} \tag{2}$$

      with $D_b = 1.87$, which is the fractal dimensionality of the percolation backbone in 3D (Hunt and Sahimi, 2017a). (Although
the flow, which supports soil development is mainly vertical, the connectivity of the dominant flow paths is three dimensional.)
      Solute advection has been shown to be more efficient than diffusion over length scales of about 10 pore separations (Hunt and
      Manzoni, 2016) allowing diffusion to be neglected in the current discussion. Further, solute transport limits chemical
      weathering and soil depths under almost any natural conditions (e.g., Egli et al., 2019).

      Predictions for flow and for crop heights can be made using (Hunt et al., 2021c)

$$x = x_0 \left(\frac{t}{t_0}\right) \tag{3}$$

      which represents a scale-independent flow velocity. This prediction was based on the hypothesis that constraints on plant
      growth resulting from a) the need to search for nutrients along optimal soil paths, b) the access of water, are eliminated by
      supplying the plants with all necessary nutrients as well as adequate water for growth.

      Eq. (1) was shown to be valid for root lateral spread of a single plant or clones at time scales of up to 100 kyr. Solving for time
instead of size yields,

$$t = t_0 \left(\frac{x}{x_0}\right)^{1.21} \tag{4}$$

      Eq. (4) is then interpreted to give $t$ as the time of formation of a self-organized clade of size x with the genetic material required
      to express a completely coupled biogeochemical cycle, such as plant material production and consumption.

      Based on the fundamental hypothesis that associations of local pores within the framework of percolation theory can generate
dominant transport paths of lowest resistance, the preceding hypotheses were used to generate the correlations between the





maximum crop height, natural vegetation RLS, and soil depth by using the same values of $x_0$ and $t_0$, the same network structure and flow properties, and exponents that were universal, as generated from percolation theory (Hunt and Sahimi, 2017a).

### 3 Experimental Data and Observations

Producing an ice age through biological innovation can be viewed as a climate crisis generated by a component of the
ecosystem, namely the plants, which robustly overproduce, whereas achievement of homeostasis afterwards would be consistent with establishment of a global-scale, adaptive "organism," or a nested system of ecosystems comprising a "symbiotic planet" (Margulis, 1999).

We propose here that the time required for a soil ecosystem to self-organize over continental scales in response to a major (e.g., photosynthetic) innovation is predictable in the context of our equations that describe soil transport phenomena. These
equations describe the two dominant natural pathways promoting communication, advective solute transport, and plant root (or fungal hyphae) extension, as well as a pathway facilitated by humans through artificial fertilization. Of these, we applied the faster of the two natural processes, the root growth model. The test is composed of two parts; 1) verification over testable time and spatial scales (up to 100kyr and 10km), 2) application to suitable events over larger scales. The biosphere's adaptation that removed the effects of the climate shocks (ice ages) brought on by colonization of the land by plants required less than
100Myr with associated continental spatial scales of 10,000km or less. Eq. (4) has been previously tested at length scales from about 10 μm to 10 km, i.e., 9 of the necessary 11-12 decades of length scale with the corresponding number of decades of time scale determined by multiplying 9 and 11-12 by $1.21 = D_{opt}$. Establishing the relevance of a scaling relationship over ca. 80% of the necessary range suggests a strong case for its continued suitability over the remaining 20%. Further, as will be seen, the scaling relationship was tested on a large neutral data base of ca. 6,000 pairs of points (Falster et al. 2015).
The experimental data used were accessed previously and include sources for soil depths as well as plant heights and root radial extents (Hunt and Manzoni, 2016; Hunt et al., 2020; Hunt, 2017; Hunt et al., 2023; Egli et al., 2018; Yu et al., 2019; Yu and Hunt, 2017ab; Hunt et al., 2021c, 2022). We note that the first of these references alone contains ca. 50 different data sources and is a source for much of the following. We analyzed three categories of data: crop height, plant height (and the equivalent root lateral spread), and soil depth. Since natural vegetation growth characteristics were assumed to be governed
by the need to find nutrients and/or water within the soil medium, plants which were heavily fertilized and watered, were considered separately and called "crops." Crop heights derived mostly from annual crops, such as beans, peas, corn, hemp, tomatoes, sunflowers, tobacco, wheat, and amaranth. On the shortest time scales, this category also included root-tip extensions (Watt et al., 2006). Because trees in plantations are heavily nourished, such as in *Eucalyptus* plantations, these were also treated in the data analysis as crops. In the case of short-term laboratory measurements of root tip extension rates, distinctions between
crops and natural vegetation were made on the basis of the description of the environment. When the environment tested had either too little water or too much salt (Watt et al., 2006), for example, such experiments were included within natural vegetation. Plants grown under conditions described as ideal were considered again as crops. The BAAD database (Falster et





al., 2015) for plant height was also divided in this fashion, with the relatively small database for fertilized *Eucalyptus* plantations added into the crops category, extending that data set to more than two years.

Climate, rather than temporal, effects on growth rates were, e.g., inferred from dominant tree heights of *Eucalyptus regnans* along a climate gradient in Australia (Givnish et al., 2014), which ranged from 4 m to 88 m where the ratio of precipitation to pan evaporation increased by a factor 20. Later results (Hunt et al., 2020, 2022) confirmed the proposed dependence of growth rates on transpiration, primarily related to climate variables, but secondarily to soil properties.

The definition of soil depth used in the data referenced was the depth to the bottom of the B (or Bw) horizon (Yu et al., 2019;
Yu and Hunt, 2017ab; Hunt et al., 2021). Egli et al. (2018) incorporated also the BC-horizon, by adding ½ of its thickness to the total soil thickness and contributed the world's largest data base of usable soil depth/age relationships. Soil depth data from deep time (over time scales from 10 Myr to 130 Myr) was described in original publications (cited by Hunt et al. (2021ac)) as "deep tropical weathering," with specific labels such as "laterite" and "saprolite," although climate regimes for some sources were humid temperate continental.

It has now become clear that the early colonization of land was by rooted plants (Morris et al., 2018; Puttick et al., 2018), though fossil evidence suggests quite short root systems. Initiated near 500 Ma (Morris et al., 2018; Pennisi, 2018), the effects of the dramatic increase in photosynthesis on atmospheric $CO_2$ content appear to have led to large-scale glaciation by 488 Ma (Lenton et al., 2012). The end of the glacial episode is estimated to be at 440 Ma (Lenton et al., 2012), 60 Myr after initial colonization. Given such a climate shock, return to equilibrium could roughly coincide with development of Margulis'
symbiotic planet.

Later colonization of the land by significantly rooted vascular plants at 420 Ma (Morris et al., 2018; Puttick et al., 2018) was followed by cooling and a glacial episode lasting approximately from 372 Ma to 359 Ma (Streel et al., 2000). The time required for reestablishment of ideal environmental temperatures for life was approximately 60 Myr since initial colonization.

## 4 Results and Discussion

The value of $x_0 \approx 1 \mu m$ was estimated (Hunt and Manzoni, 2016) as a fundamental pore scale, while the time scale $t_0$ was taken as the time for water flow across a 1μm pore at a typical subsurface flow rate and was calculated as $t_0 = 1\ \mu m\ /\ 1\ \mu m\ s^{-1} = 1$ s). This prediction is given by the dashed red line in Figure 1 and compared with actual woody plant data. For lengths of centimeters or less, the data reflect laboratory root tip or fungal hyphae extension rates. On scales exceeding 120 m, data reflect the RLS of single organisms (clones) with multiple subaerial stems, both plants and fungi (Hunt et al., 2021c; Arnaud-Haond
et al., 2012), while at intermediate length scales, the data are for vegetation height, which is known to be nearly equivalent to RLS on length scales between about 0.5 m and 40 m (Hunt and Manzoni, 2016). For $x > 10$ km, we associated the interpretation of an organism with the idea of Lovelock's Gaia, as modified by Margulis.

Figure 1 indicates near conformance of Eq. (4) with the upper bounds of the RLS data on time scales ranging from minutes to 100 ka, and length scales from 100 μm to 10 km. Using the same equation, the time for a continental scale (5,000 km)
"organism" to develop was estimated at 80 Ma. Although the slope of the line is universal, the value of the time scale required



for an "organism" of 5,000 km changes, if the parameter values for Eq. (4) are different. Later, (Hunt et al 2020; 2022; Yu and Hunt, 2018; Egli et al. 2019, etc.) $x_0$ was defined explicitly as a median particle size or plant xylem diameter, for which 10 μm is a reasonable estimate (Watt et al., 2006). Regardless, particularly for vegetation, the relatively small variability in the choice of the fundamental length scale has insignificant effect on the predicted time for a continental scale to be reached, although

the prediction of Eq. (4), through the factor $t_0$, is quite sensitive to the fundamental flow rate.

The close agreement between predicted and observed soil depths and root lateral spreads as well as vegetation heights has been discussed elsewhere. The prediction, that it requires 80 Myr for continental-scale adaptation to new conditions according to scaling relationship, Eq. (4), is only 33% larger than the 60 Myr value, as inferred from recovery from each of two Paleozoic ice ages triggered by plant innovations. Note that the prediction of 80 Myr was given already in Hunt and Manzoni (2016);

what is new here is the specific application to Paleozoic ice ages. This reasonably close correspondence suggests that looking for universal time scales of establishing steady-state climatic conditions using universal scaling of vegetation and fungal growth rates has potential for helping to understand past climate fluctuations.

That the scaling relationship in Eq. (4) for coupled plant/fungus/bacterial spread could be extended to planetary scales as a means for understanding Gaia-like adaptation to land colonization by plants would be in accord with Lovelock's comment in

a relatively recent interview (Ball, 2014) in Nature: "I'm very intrigued by the latest attempt to resuscitate the idea that all of climate regulation is done by rock weathering. The geologists keep on ignoring the bacteria." The research of Hunt and Manzoni (2016) discussed here and elsewhere is entirely consistent with Lovelock's implication. First, rock weathering rates in situ do not depend directly on temperature, but on moisture fluxes (Hunt, 2017; Egli et al., 2018; Yu et al., 2019; Yu and Hunt, 2017ab; Hunt et al., 2021ac; Hunt and Sahimi, 2017b), which restricts the efficacy of the rock weathering thermostat to

periods of time when atmospheric moisture and temperature are in phase. When high temperatures and atmospheric $CO_2$ content are not in phase with greater precipitation, e.g., during the great Permian extinction (a likely result of unification of Earth's land masses in Pangaea under relatively arid conditions) (Hunt and Sahimi, 2017b), the rock thermostat fails and temperatures remain high. Second, although the solid inorganic carbon reservoir volume is much larger than the organic carbon reservoir, Figure 1 demonstrates clearly that, particularly changes in its rate of storage, are much slower because of the much

lower slope produced by the exponent $1/D_b = 1/1.87 = 0.53$ as compared with $1/D_{opt} = 1/1.21 = 0.83$. Specifically, at 100 kyr, the predicted and observed soil/weathering depth is at most a few meters, while the predicted and observed laterally integrated biological dimension reaches 10 km. Any pore-scale horizontal advection driven by plants is of a similar magnitude to the vertical advection driven by gravity, since the magnitude of the evapotranspiration flux is typically nearly 2/3 of the infiltration flux (Hunt et al., 2023); thus spread of, and communication between, bacteria in the subsurface can be considerably enhanced

in the horizontal directions (relative to vertical) through coupling with plant processes. Also, consistent with the original suggestion (Hunt and Manzoni, 2016) that the required global "organism" could be a plant/bacteria soil ecosystem (in which plant matter breakdown and production are in steady-state), homeostasis occurs when a global balance in $CO_2$ drawdown and emission is achieved at the right atmospheric composition that a moderation of temperature is achieved. Here such homeostasis emphasizes the key roles of bacteria and fungi, but allows their spread and adaptation to be controlled by the underground



spreading rates of plant roots and fungal hyphae, a coupling to the plant ecosystem that ensures similar time scales of the spread of adaptations for plants and bacteria at all length scales. The scaling relationships, on which this manuscript is based, thus tend to emphasize the relevance of biota, including bacteria, in developing a large scale, restorative response, to climate perturbations, in comparison to the abiotic (rock cycle) response, which is not necessarily identifiable as a negative feedback. The same scaling relationships used here also form the basis for an accurate (within 1.5%) prediction of the global fraction of

precipitation returned to the atmosphere through terrestrial evapotranspiration (Hunt et al., 2023). The result is based on maximization of plant productivity using a thin root layer defined by the soil depth, and a global-scale assumption of neither energy nor water limitations (Hunt et al., 2023). Moreover, the associated dependence of the water balance (Hunt et al., 2023) on climatic variables was recently replicated in a dynamic process model (Nijzink and Schymanski, 2022), as shown by Hunt et al. (2023). Thus, although a soil ecosystem may promote homeostasis, plants by themselves fit a model of "greed" with

respect to $CO_2$, taking as much as they can, in order to exploit solar and soil resources as possible to most effectively cover Earth's surface with plant matter. Taken together, our results imply that negative feedback for climate change is generated in the soil and not through plant interaction with atmospheric chemistry, in accord with cited evidence (Kirchner, 2002) that a modern 25% increase in atmospheric $CO_2$ content led to only a 2% increase in productivity. Productivity is more directly tied to (evapo-) transpiration, which is enhanced at higher temperatures. Considered together, these implications remove an

additional criticism (Kirchner, 1989, 2002) of the homeostasis hypothesis, namely that Gaia cannot simultaneously be expected to moderate increases in solar irradiance (a one-way feedback) and stabilize the temperature against fluctuations (a two-way feedback). For it is the plants acting alone that produce the first effect, but the coupling of the plants and the bacteria that produce the second.

The question of whether life on our planet tends to promote climate and chemical homeostasis (Gaia), or to cause fluctuations

that endanger life (Medea), is of fundamental importance to understanding Earth's history as well as the future of our species. We do not answer this question, but address specific criticisms of the Gaia hypothesis. Through a kind of conceptual division and synthesis of ecosystems into their parts and their reassembly, we can identify the component - "greedy" photosynthesizers - that has led to climate shocks, and the whole that has restored equilibrium. Especially, we believe that the result reported here, regarding recovery from the Paleozoic ice ages, can help point the way to a future in which the opposite sides of the Gaia

hypothesis can coexist meaningfully and usefully.

## 5 Conclusions

An existing prediction for the development of a self-organized network of interacting ecosystems to reach continental spatial extent and that can promote climate regulation by producing a steady-state biogeochemical cycle based on the production and consumption of plant material was tested on the climate shocks produced by two land plant innovations that generated ice

ages. The prediction was generated by a percolation-based scaling relationship for plant root and fungal hyphae growth rates in the subsurface. It was suggested that opportunistic bacteria would also take advantage of this (relatively rapid) means of dispersal. The explicit comparison was to the time required for the biosphere to adjust to the subsequent ice ages. These



parameter-free predictions were each verified to within 33%. For a process that links the pore scale of less than one minute with large intervals of Earth's history (60Myr), the success of this prediction appears to exceed usual expectations.

We believe that taken together with other results, the present comparison implies that: 1) The scaling of transport times with distance is a critical input to understanding critical zone science, particularly at larger spatial and temporal scales, 2) This scaling is not, in general, linear, 3) Exponent values can be predicted by percolation exponents, 4) Non-linear hydrologic, geochemical, and ecological scaling predictions and optimizations link these three disciplines in the critical zone, 5) Additional support for a weaker Gaia hypothesis in line with Doolittle's (2017) Darwinization of the Margulis (1999) emergent behaviour

of coupled ecosystems based on the long-term stability of clades that integrate biogeochemical cycles is found.

**Data availability**

Data used in this study are available at: http://www.hydroshare.org/resource/eef6b82c0eaa48e78b68b752da2b4ba5

**Author contributions**

A.H.: Data curation, Conceptualization, Methodology, Formal analysis, Validation, Roles/Writing - original draft; Writing - review & editing. M.S.: Conceptualization, Methodology, Validation, Roles/Writing - original draft; Writing - review & editing. B.F.: Conceptualization, Validation, Roles/Writing - original draft; Writing - review & editing. M.E.: Methodology, Validation, Roles/Writing - original draft; Writing - review & editing. Z.J.K.: Methodology, Validation, Roles/Writing -
original draft; Writing - review & editing. B.G.: Data curation, Formal analysis, Validation, Roles/Writing - original draft; Writing - review & editing. F.Y.: Methodology, Validation, Roles/Writing - original draft; Writing - review & editing.

**Competing interests**

The authors declare no competing interests.




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



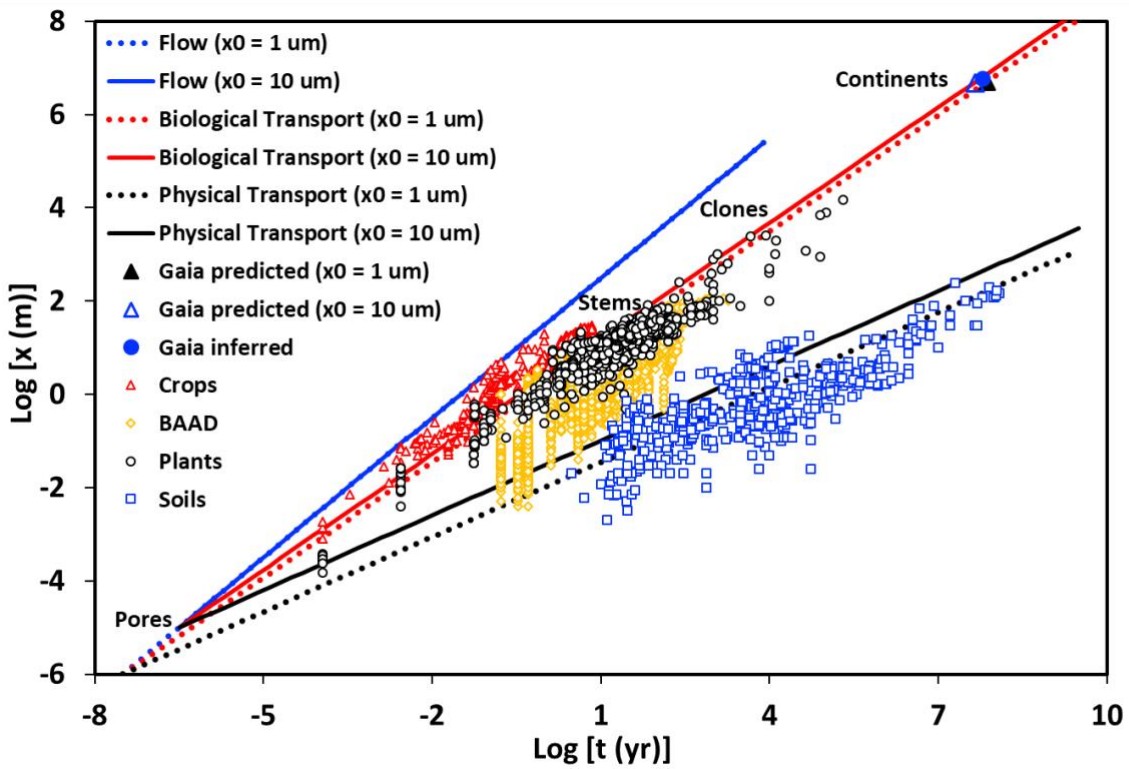

**Figure 1: Application of scaling relationships to biological and physical transport processes, assuming a flow speed independent of scale (blue) with value of about 1 μm s⁻¹ taken from a summary (Blöschl and Sivapalan, 1995). Root**

**growth rates (red) decline over time according to the 2D optimal paths exponent from percolation theory (Eq. (4)), soil formation rates (black) decline according to the scaling of solute transport using the fractal dimension of the percolation backbone. "BAAD" refers to the Falster et al. (2015) database, "plants" were from many sources (Hunt and Manzoni, 2016; Hunt, 2017; Watt et al., 2006; Arnaud-Haond et al., 2012), selected for faster growth, and "soils" are soil depths (data compilation (Hunt and Manzoni, 2016; Hunt, 2017; Egli et al., 2018; Yu et al., 2019; Yu and Hunt, 2017ab; Hunt**

**et al., 2021c)). Dashed lines reflect predicted times using the originally suggested length scale of 1 μm, while solid lines indicate the changes resulting from revising x₀ to 10 μm, a change scarcely visible for predictions of plant growth.. "Gaia predicted," 80 Myr, is the time for Eq. (4) to generate 5,000 km, as originally predicted. "Gaia inferred" pairs a physical extent of ca. 6,000 km, obtained as half the square root of the area of all the continents, with the time for emergence from an ice age (60 Myr).**
