# Peer review of "Gaia: Complex systems prediction for time to adapt to climate shocks"

_Earth System Dynamics, 2023_

## Author Comment (AC1)

Our responses to Dr. Manzoni's comments are in red.

Hunt and co-authors propose a new hypothesis to explain the duration of glaciations during the period 500-350 million years ago, based on the characteristic velocity of root and hyphal spread in soil. They argue that roots and hyphae (and associated bacteria) would spread over continental scales, thus transporting genetic information (and evolutionary adaptations), with a characteristic time scale comparable to those typical of the glaciation durations in that period. This would indicate that these components of the ecosystem cooperate to stabilize climate even if plants by themselves would tend to destabilize it due to their efficient uptake of atmospheric $CO_2$ (which in turn cools climate). Cooperation would work because fungi and bacteria would co-evolve with plants to efficiently decompose organic matter, returning $CO_2$ to the atmosphere. I hope I understood the proposed rationale correctly, because it is not described very clearly.

This hypothesis is tested using a relation between root or hyphae extension as a function of time derived from 2D percolation in porous media. The idea is that roots and hyphae extend in soil in an 'optimal' way, so that the distance travelled scales as a power law of time with a predictable exponent from percolation theory. This relation is consistent with the rate of growth of plants in optimal growing conditions, which in turn is proportional to the rate of root growth, lending some support to the theory. However, this theory was developed at individual plant scale, extending at most to clonal plants or hyphal networks spreading over ~100s of meters (or few km for some fungal networks). Here the authors extend this concept to the continental scale, resulting in time scales ~60 million years.

My main concern is at a conceptual level—evolutionary innovations do not need to be transported through the soil because they can spread orders of magnitude faster by other media. Spores and seeds of plants, and spore (or entire cells) of microorganisms can be transported by wind or animals, or via eroded soil in surface water bodies. Underground dispersal might be used, but over short distances, and it seems a bit far fetched to hypothesize that it is a relevant mechanism over continental scales. One could even argue that it would take a single river or a mountain chain to stop entirely this mode of dispersal across a continent. Anything stopping roots and hyphae would drastically hamper transport underground. Moreover, from an evolutionary perspective, solutions that are costly are outcompeted—and growing roots and hyphae is a costly way to colonize new land.

My other general comment is that it is not clear which adaptations would be transported. I would present evidence of co-evolution of say lignin-based wood and ligninolytic enzymes to develop arguments of coordination between plants and saprotrophs. But even with strong evidence of such a co-evolution, I am not sure it would be possible to support the proposed hypothesis given the other (faster, metabolically less costly) dispersal modes available to producers and decomposers.

Dr. Manzoni expressed concerns that 1) evolutionary innovations may be *spread orders of magnitude faster* by pathways other than through the soil, 2) soil pathways are more costly metabolically, and 3) soil pathways can be interrupted by geographic features such as rivers or mountains, 3).

At short time scales under modern conditions, diffusion of seeds by, e.g., birds, is so rapid that such arguments, when applied to single species over short length scales, are

undoubtedly valid. A common model of seed dispersal is diffusion and a cursory investigation of reported model parameters (https://www.jstor.org/stable/pdf/2261103.pdf) reveals that the relevant diffusion coefficients are on the order of, e.g., $25 m^2/yr$. But, application of such arguments to the distant past may be based on biases from the present. We suggest several caveats should be considered that, when taken together, can change the relative importance of soil-based and atmospheric mechanisms. First, it should be noted that advection-based mechanisms, such as the soil growth mechanism that we proposed, gain in speed relative to diffusion when the spatial scale increases. From 10's of meters to 10,000 km represents 6-7 orders of magnitude variations, which is probably insufficient for the present conditions to provide sufficient slowing of diffusion for domination of our root-growth related mechamism, but which is likely to be relevant to Paleozoic conditions. For example, the calculation referenced above included individual time steps controlled by the advection from flying birds over a period of time equal to that required for the seed to pass through the gut. Birds and seeds were, of course, absent in the Paleozoic. Second, the overall conditions for plant growth were very different in the era before the plants modified both soil and atmosphere (Kleidon, 2002) to their advantage. Third, it is necessary to transport the entire clade, not just a single species. Fourth, we can appeal to the dates already mentioned to suggest that plant dispersal in this time period was not orders of magnitude faster than the relaxation to homeostasis. After the initial invasion of the land by plants, it took at least 12 million years before initiation of an ice age (from 500 Ma to 488 Ma), and after the second wave of innovation starting at 420Ma, 48 million years (until 372 Ma). These two results support an inference that attaining homeostasis requires time scales between 5 times as long and 5/4 as long as the full exploitation of land-accessible resources by plants. Nevertheless, if diffusion were not a noticeably faster mechanism, at least over smaller length scales, one would not expect that interruptions by rivers could be overcome by a faster mechanism of dispersal operating in series with the slower "through the soil" mechanism postulated here. Under these circumstances, we do still suggest that the close correspondence of a parameter-free prediction with the observed time scale warrants serious consideration as a viable hypothesis.

The evidence on the time taken after the initiation of colonization of the soil by plants before the initiation of an ice age does not appear to support Dr. Manzoni's statement on long time scales, however. For colonization at 500Ma, continental glaciation followed at 488Ma, and for the later plant colonization at 420Ma, glaciation followed at 372Ma. Thus, the first case quoted implies plant spread at 5 times the rate at which homeostasis was obtained, but the second case quoted appears to involve only a factor of 1.25 distinction. That is indeed evidence that plant spread is more rapid than the time scales predicted purely through subsurface plant growth, and may help understand Dr. Manzoni's second concern as well; blockage by rivers may hinder root spreads at small spatial scales, but if diffusion of genetic information over short distances through other mechanisms, such as seed dispersal, tree-throw, animal transport, etc. is faster, then the combined mechanism would operate relatively unimpeded, but with the time scales of the root growth.

Other comments

L91: I would not agree with this strong statement (see my comment above)

We can always modify strength of any statement.

L129: just a detail—aren't data and observations the same thing?

Observations are often converted to reported "data" through inverse modeling, such as soil ages from radionuclide data. The original observations are mostly data, but what is reported is not always.

L135: how do human improvement of plant growing conditions fit in this work? We are dealing with deep past when conditions were probably far from optimal in many areas of the world

Plants tend to optimize conditions for their growth, where possible. Maximum biomass production by an ecosystem, gives it access to as much light energy as possible, and is what has allowed prediction of the water balance and associated streamflow (see https://eos.org/editor-highlights/how-much-terrestrial-precipitation-is-used-by-vegetation) and Axel Kleidon to point out that soil and atmospheric conditions with plant modifications are more conducive to plant growth than without.

L141-142: argument is not clear

Will try to clarify in a revised version.

L184: "subaerial stems" meaning roots and hyphae?

This was mainly to address the fact that the root systems of clonal plants are connected and new stems can grow from any portion of that root system. Probably this particular phrasing should be adjusted for better communication.

L192: plant xylem is not a random medium, so I am not sure percolation theory can be applied as in soil

Actually, some aspects of percolation theory can be applied anywhere, such as critical path analysis. Nevertheless, in the case that the medium is not random, or that it is highly correlated, there is likely no particular advantage in applying critical path analysis, unless one is addressing a suite of media that can vary across the entire spectrum of disorder (see an article in JGR by Bernabe and Bruderer, 1998, comparing Kozeny-Carman, stochastic methods, and critical path analysis in determining an effective hydraulic conductivity. To what degree this might apply in a plant xylem is unknown.

L203-213: I cannot follow this argument—how are the time scales of horizontal transport and weathering connected? My understanding was that the proposed hypothesis was not about weathering, but about producers and decomposers finding a 'balance' to recycle CO2 and keep the Earth warm.

That is indeed our suggestion; however an alternate suggestion is that rock weathering is the thermostat, not biota. This should not be sprung on the readers as we did, but introduced more properly.

L229-234: also here I find it difficult to follow the presented arguments—what is the connection with the proposed hypothesis?

We will try to clarify in a revision.

L238: gross primary productivity varies with increasing atmospheric CO2 in the range 1-6 gC/m^2/y/ppm (https://doi.org/10.1073/pnas.2115627119). Thus, a 25% increase in CO2 (100 ppm) leads to an increase ~100-600 gC/m^2/y, which is much more than the 2% figure reported. Additional explanations are needed here, including recent estimates of productivity sensitivity to CO2

We will address in any revised manuscript.

L238-239: productivity is related to transpiration rate (only a fraction of evapotranspiration), but when changing temperature, the relation might break down as warmer conditions (for given water vapor content) trigger stomatal closure and decrease photosynthetic rates

Currently, the global average of the ratio of transpiration to evapotranspiration is 2/3. Stefano is correct that this average might not remain constant with changing climate; the mean will not be observed everywhere under all conditions, so it is difficult to predict how the distribution of the values of this ratio would behave. We will not attempt to, but we will acknowledge the uncertainty in a revised version.